# Longitudinal association between self-rated health and psychological well-being in a sample of Spanish university graduates

Maria Vasilj[1], Estefanía Toledo[1,2], Prado Silván-Ferrero[3],
Encarnación Nouvilas-Pallejá[3], Alessandra Jarufe[4], Leticia Goni[1,2],
Maira Bes-Rastrollo[1,2], Miguel Ruiz-Canela[1,2]*

1 University of Navarra, Department of Preventive Medicine and Public Health, IdiSNA (Instituto de Investigación Sanitaria de Navarra), Pamplona, Spain, 2 CIBER Fisiopatología de la Obesidad y Nutrición (CIBERObn), Instituto de Salud Carlos III, Madrid, Spain, 3 National University of Distance Education, School of Psychology, Madrid, Spain, 4 Pontificia Universidad Católica de Chile, Santiago, Chile

* mcanela@unav.es

## Abstract

### Purpose

Previous research has shown that psychological well-being (PWB) is associated with self-rated health (SRH) over time, but limited research has explored the reciprocal association. This study examined the longitudinal association between self-rated health and subsequent psychological well-being in a Spanish cohort.

### Methods

Data were collected from participants of the "Seguimiento Universidad de Navarra" (SUN) Project, a dynamic university graduate cohort study. Psychological well-being was measured using the Ryff scale. Multivariable linear regressions analyse assessed the relationships between self-rated health, measured at study's baseline and after 4 years, and psychological well-being measured after a mean follow-up of 14.6 years. The association with Ryff's six dimensions (self-acceptance, positive relations with others, autonomy, environmental mastery, purpose in life, and personal growth) was also assessed.

### Results

A total of 2,927 participants, mean age 38.7 years (SD = 11.2) at pre-baseline, 55.5% women, were followed-up for a mean of 14.6 years (SD 0.7). Compared to participants reporting poor/fair health, those with excellent self-rated health at baseline showed substantially higher subsequent psychological well-being scores (ajusted β = 15.1, 95% CI: 11.4–20.3) and were three times more likely to have above-median psychological well-being (adjusted OR=3.0, 95% CI: 1.8–4.8). A dose-response relationship was

**Data availability statement:** Data available from 10.6084/m9.figshare.29278433.

**Funding:** This work was supported by the Spanish Government–Instituto de Salud Carlos III, Fondo Europeo de Desarrollo Regional (FEDER), co-funded European Union: PI23/01332, PI24/0173, and the Navarra Regional Government (19/2023).

**Competing interests:** The authors have declared that no competing interests exist.

observed across self-rated health categories, with significant associations for all PWB dimensions. Additionally, self-rated health improvements during the first 4 years were associated with higher subsequent psychological well-being scores.

## Conclusions

Self-rated health showed a direct, long-term association with subsequent PWB among Spanish university graduates, suggesting that perceived health status may be an important determinant of future PWB. However, our findings are limited by the single measurement of PWB and the use of self-reported measures. Further studies in more diverse populations are warranted to confirm these results.

## Introduction

Psychological well-being (PWB) is a state of happiness and contentment and not merely the absence of mental health issues [1–3]. Previous research has explored how different health conditions and illnesses, such as heart disease, different types of cancer and frailty, might be related to PWB [4–9]. Different health behaviors such as exercise and sleep, have also been linked to PWB [10–13]. While these results provide valuable insights into the impact of various physical health circumstances and habits on PWB, there is a gap in the literature regarding studies that examine overall health in relation to PWB.

Self-rated health (SRH) is one of the most widely used indicators of overall health. This single question [14] has proved to be a simple and reliable measure of the general health status [14] and to predict mortality [15,16], decrease in functionality [17], and use of health services [18]. SRH has also been shown to be associated with psychological disposition [19], positive and negative affect [20], and quality of life [21]. However, research examining the relationship between SRH and PWB remains limited and methodologically constrained [22–28]. Existing studies are predominantly cross-sectional [22,24–27] or feature short follow-up periods, limiting causal inference. Moreover, available longitudinal research has primarily investigated PWB as a predictor of subsequent SRH [23,28], with Ryff et al. demonstrating that higher PWB predicted better self-rated health over time [23], and Choi & Jung's machine learning analyses identified PWB as a strong predictor of SRH [28]. This unidirectional focus leaves a significant gap in understanding whether SRH might similarly influence future PWB. More critically, this relationship remains largely unexplored in longitudinal designs. Therefore, investigating the prospective association between SRH and subsequent PWB over an extended follow-up period is essential to gain a more comprehensive understanding of the bidirectional relationship between these constructs.

SRH is not a static measure but rather can vary over time in response to various life circumstances and experiences. Longitudinal studies have shown that SRH is influenced by multiple factors including changes in health behaviors, physical and mental health status, social relationships, and socioeconomic conditions [29–32]. These temporal changes in health perceptions may differentially affect subsequent psychological well-being compared to baseline health assessments alone. Therefore,

examining changes in SRH over time provides a more dynamic understanding of how changes in perceived health influence long-term psychological outcomes.

According to Ryff [33], PWB is composed of six dimensions: self-acceptance, positive relations with others, autonomy, environmental mastery, purpose in life and personal growth. This theoretical framework has been tested in different populations, and comparative studies with other models and theories have been conducted [34–36]. In this study, our hypothesis was that better SRH will be positively associated with overall PWB and its individual dimensions. The primary aim was to evaluate the longitudinal association between SRH and PWB, as defined by Ryff's six dimensions, within a cohort of Spanish university graduates after a mean follow-up of 14.6 years (SD 0.7). As a secondary objective, we examined whether a 4-year change in SRH was associated with PWB.

## Methods

### Design

The Seguimiento Universidad de Navarra (SUN) Project is a prospective and dynamic cohort study, composed of university graduates. Further details about the SUN Project have been previously published elsewhere [37,38]. Briefly, participants' recruitment started in December 1999, and remains permanently open. Participants are invited through alumni associations and professional associations (such as the Alumni Association of the University of Navarra and regional associations of physicians, nurses, pharmacists, dentists, and engineers) throughout Spain. The first extensive questionnaire, consisting of 554 items, is implemented to collect socio-demographic variables, lifestyle factors, health-related data, and dietary information. Subsequent shorter questionnaires are sent every 2 years for follow-up, including items on changes in lifestyles and incident diseases. All questionnaires are sent either by mail or email.

For the present analysis, we included participants who reached the 14-year follow-up questionnaire by May 2022. Data for this analysis were accessed from the SUN Project database version dated September 23, 2022.

Fig 1 shows the time points used to collect data for the present study (t0, t1, and t2). First, covariates were measured during the first time point (t0), defined as the pre-baseline wave. Second, SRH, the principal independent variable, was assessed at the baseline wave (t1) and at year 4 of follow-up. Third, PWB was measured at the outcome wave (t2).

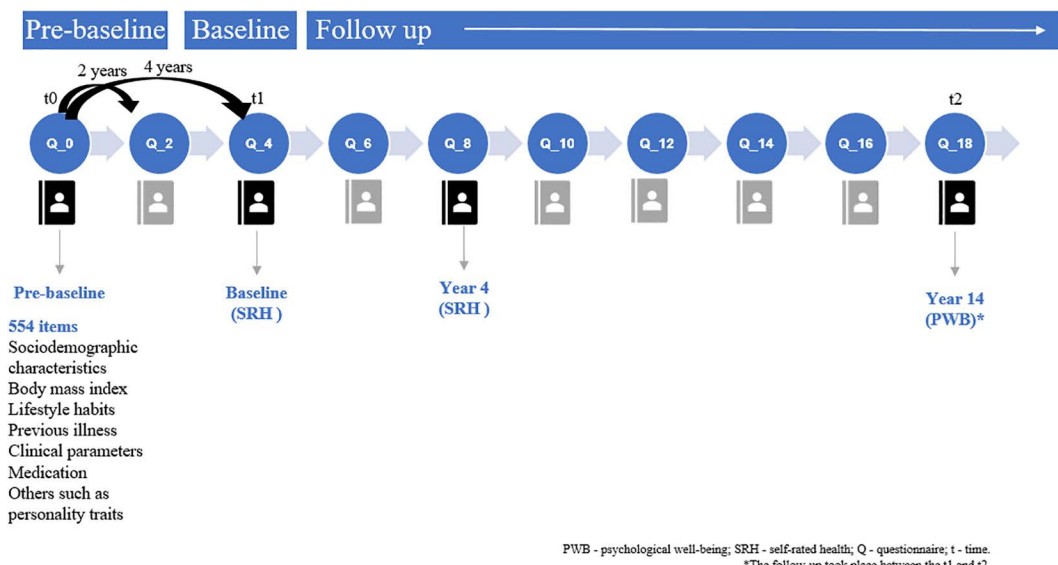

**Fig 1. Methodology of the SUN cohort (data collection for the present study).**

## Participants

By May 2022, 3 112 participants had completed the 14-year follow-up questionnaire (t2). For the present analysis, the sample was restricted to individuals with (1) follow-up from pre-baseline until t2, (2) who completed the Ryff's PWB scale, and (3) answered the self-rated health question at baseline (t1). This resulted in a final sample size of 2 927 participants (Fig 2).

At pre-baseline, participants received written information on their specific data to be requested by future questionnaires, the protection to safeguard their privacy, and the future feedback from the research team. All potential candidates were informed of their right to refuse to participate in the SUN study or to withdraw their consent to participate at any time without reprisal, according to the principles of the Declaration of Helsinki. The voluntary completion of the baseline questionnaire was considered to imply informed consent. The Research Ethics Committee of the University of Navarra approved this method to request the informed consent of the participants.

## Variables

**Sociodemographic characteristics, health-related factors and lifestyle habits.** At t0 (pre-baseline time), a 554-item questionnaire was used to collect data on health status, prevalent diseases, medication, lifestyle factors, food intake, personal traits, etc. Sociodemographic variables (sex, age, marital status, number of children, and educational level), health related

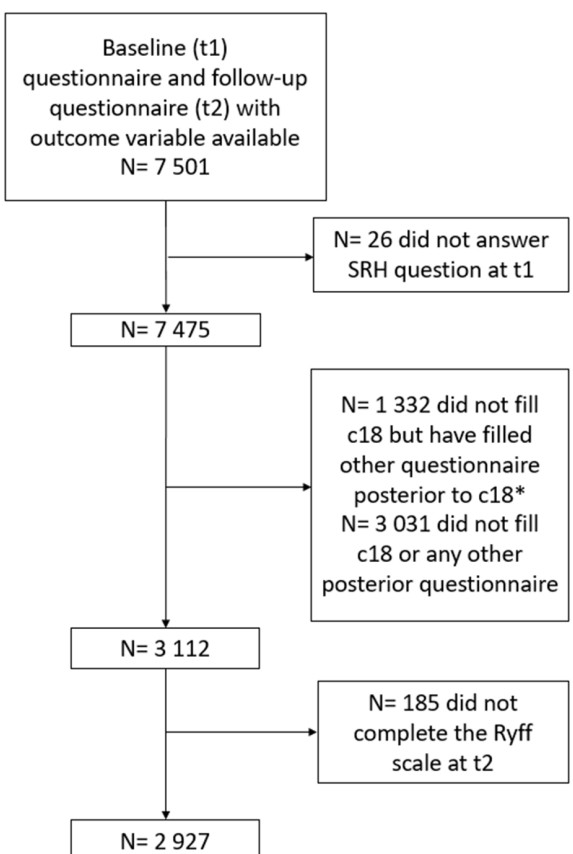

**Fig 2. Flowchart of inclusion criteria.** * Some participants skipped certain questionnaires but completed later follow-up assessments. These participants are not considered lost to follow-up, instead they exhibit inconsistent response patterns throughout the study period.

factors (prevalence of depression, diabetes, cancer, cardiovascular disease, and body mass index (BMI)) and lifestyle habits (diet, physical activity, sleep, alcohol and tobacco consumption) were measured in the initial questionnaire at t0 (Fig 1). In the same questionnaire, participants also answered a question about the frequency with which they socialized with friends. This variable was used as an indicator of social interaction frequency and has been widely applied in epidemiological research examining the associations between lifestyle habits and both mental [39–41] and physical health outcomes [42–45].

**Self-rated health (SRH) assessment.** To measure SRH, we used a single item rated on a 5-point Likert scale. Participants had to respond to the question "In general, you would say that your health is…?" using answer categories between "excellent" and "poor" (i.e., 1 = excellent, 2 = very good, 3 = good, 4 = fair, 5 = poor). Although self-report questionnaires may introduce potential bias, the validated nature of our instruments and the high educational level of participants likely enhanced response reliability and data quality. In order to facilitate the interpretation of the results, we inverted the scoring for this question so that a lower score indicated worse SRH. Since very few participants rated their health as poor, they were classified into 4 categories: fair or poor, good, very good, and excellent health. In our study, this question was included in the questionnaire Q4, which corresponds with the baseline time point (t1) and in the questionnaire Q8, which corresponds with year 4 after baseline (Fig 1).

**Psychological well-being (PWB) assessment.** PWB was measured at questionnaire Q18 (Fig 1). The Spanish adaptation of Ryff's six dimensions for autonomy, environmental mastery, personal growth, positive relations with others, purpose in life, and self-acceptance was used [33,46]. This version of Ryff's dimensions contains 29 items in total, with each one consisting of four to six items, with a mix of positive and negative items. Participants indicated the extent (1 = strongly disagree; 2 = quite disagree; 3 = somewhat disagree; 4 = somewhat agree; 5 = quite agree; 6 = strongly agree) to which the statements described them. Negative items were reverse coded so that higher scores reflected a more positive evaluation. The total score of PWB was obtained by summing up the results of all 6 scales. The minimum observed score was 41 and the maximum 174.

## Statistical analysis

Multivariable linear regression models were performed to evaluate the relationship between SRH and PWB, with the fair-poor health as a reference group. A crude model was initially constructed, followed by a model adjusted for sex, age, educational level, marital status, and number of children. A second multivariable model was additionally adjusted for lifestyle habits including leisure time physical exercise (METs-h/week) [47], a score of total physical activity [48], energy intake (kcal/day), adherence to the Mediterranean diet (14-point MEDAS) [49], sleep (h/day), smoking (never, former or current smoker), lifetime tobacco exposure (pack-years), alcohol consumption (g/day), and frequency of interaction in social networks (> 3 h/week). A third model was included, incorporating the prevalence of depression (yes/no), diabetes (yes/no), cardiovascular disease (CVD) (yes/no), and cancer (yes/no), and BMI (≤25 vs >25 kg/m$^2$). Regression coefficients and their 95% confidence intervals were calculated for the overall PWB and each of its 6 dimensions based on the level of SRH. To test for linear trend, we conducted an additional analysis using SRH as a continuous variable in the regression model. To assess potential multicollinearity among independent variables in our regression models, we calculated Variance Inflation Factors (VIF).

To assess the robustness of our findings, we conducted sensitivity analyses by 1) excluding prevalent cases of diabetes, CVD, and cancer, 2) excluding prevalent cases of depression, 3) excluding all chronic diseases (diabetes, CVD, cancer, and depression), 4) by transforming the dependent variable using the inverse normal transformation to create normally distributed variable, and 5) adjusting additionally for the change in the SRH between baseline and year 4 of follow up. Participants were placed into three categories depending on whether their SRH decreased, stayed the same or increased. The same adjusted models previously described were applied.

We also stratified the results by sex, age (<40 or ≥40 years), being married (yes/no), education level (3–5 years or 6–9 years), and level of physical activity (total score ≤4 vs > 4) [48]. We assessed the potential effect modification of these

variables on the association between SRH (two categories: excellent/very good and good/fair/bad) and PWB. Product terms between these variables and SRH were calculated. The p value for interaction was obtained with the likelihood ratio test.

We used logistic regression models to assess the association between SRH and PWB as a dichotomous variable, using the median as the cut-off point. We adjusted for the same variables described before in the linear regression analyses.

Finally, to assess whether the 4-year change in the SRH was associated with PWB, we repeated the analyses adjusting additionally for the change in the SRH between the baseline and 4 years of follow up. The participants were classified into three categories: "decrease", "no change", and "increase", according to the change in their SRH over the 4 years.

Missing data was found for the following variables: marital status (0.5%), number of children (2.6%), smoking status (0.3%) and hours of sleep (17%). We assumed that missingness in smoking status and number of children was not random. Specifically, when participants do not provide information on smoking or children, it is often because they perceive the question as not applicable to them. This type of non-response is consistent with a missing not at random mechanism, as the probability of missingness is directly related to the unreported value itself (e.g., a non-smoker omits smoking status, or a participant without children omits number of children). Therefore, we coded missing values in smoking as 'never smoked,' and missing values in number of children as 'none.' For marital status, we categorized missing values as 'Other.' For hours of sleep, which showed the highest missing proportion, we used regression imputation [50] with predictors including psychological well-being (PWB), self-rated health (SRH), sex, age, physical activity, and total physical activity score. To ensure that these assumptions did not bias the findings, we additionally performed a complete-case analysis excluding individuals with missing values. Internal consistency of the Ryff's scale was assessed using Cronbach's alpha. All tests were conducted using the statistical program Stata 17, and a P-value < .05 was considered as statistically significant.

## Results

### Descriptive statistics

A total of 2 927 participants had a mean follow-up of 14.6 years (SD 0.7). From them, 55.5% were women and the mean age at pre-baseline was 38.7 years (SD 11.2). Table 1 summarizes the characteristics of the participants according to four categories of SRH. Those who rated their health higher were younger, single, had higher leisure time physical activity and total physical activity score, were more likely to have never smoked and spent more time with their friends (Table 1).

In our study, the full scale showed good internal consistency (Cronbach's alpha = 0.90). Cronbach's alphas for each individual subscale were as follows: self-acceptance = 0.78, autonomy = 0.68, positive relations with others = 0.76, environmental mastery = 0.67, purpose in life = 0.80, and personal growth = 0.66.

Multicollinearity assessment using Variance Inflation Factors revealed no problematic collinearity among covariates (all VIF values <2.0 for continuous and dichotomous variables), supporting the validity of our multivariable models.

### Main analysis results

We found better subsequent PWB in participants with good SRH ($\beta = 7.3$, 95% CI: 4.1,10.5), very good SRH ($\beta = 12.3$, 95% CI: [9.1,15.6]), and excellent SRH ($\beta = 15.1$, 95% CI: [11.4,20.3]) compared to poor or fair SRH (Fig 3 and S1 Table). A higher SRH was also related to higher scores in all 6 PWB dimensions (Fig 4 and S1 Table). These linear associations were statistically significant in the crude and multivariable models with p values for trend < 0.001.

In the stratified analyses by sex, age (<40 vs ≥ 40 years), years of education at the university (3–5 vs 6–9), marital status (married, yes/no), and level of physical activity (total score ≤4 vs > 4), slight differences were observed between the categories, but no statistically significant modification effect could be found (Table 2).

**Table 1. Baseline characteristics of participants according to categories of self-rated health: The Seguimiento Universidad De Navarra (SUN) cohort.**

| | Fair or poor (1-2) | Good (3) | Very good (4) | Excellent (5) |
|---|---|---|---|---|
| N | 119 | 1 336 | 1 201 | 271 |
| Age (years), mean (SD) | 42.5 (10.2) | 40.3 (11.2) | 37.1 (10.8) | 36.5 (10.8) |
| Sex, n (%) | | | | |
| Female | 69 (58.0) | 733 (54.9) | 677 (56.4) | 147 (54.2) |
| Marital status, n (%) | | | | |
| Single | 40 (33.6) | 475 (35.5) | 531 (44.2) | 122 (45) |
| Married | 71 (59.7) | 806 (60.3) | 636 (53) | 140 (51.7) |
| Other | 8 (6.7) | 55 (4.1) | 34 (2.8) | 9 (3.3) |
| Number of children, n (%) | | | | |
| 0 | 52 (43.7) | 620 (46.4) | 644 (53.6) | 140 (51.6) |
| 1-2 | 48 (40.3) | 486 (36.4) | 383 (31.9) | 85 (31.4) |
| 3 or more | 19 (16) | 230 (17.2) | 174 (14.5) | 46 (17) |
| Years of university education (SD) | | | | |
| 3–5 years | 102 (85.7) | 1,108 (82.9) | 973 (81) | 216 (79.7) |
| 6–9 years | 17 (14.3) | 228 (17.1) | 228 (19) | 55 (20.3) |
| Leisure time physical activity (METs-h/week) | 20.6 (27.1) | 19.7 (21.9) | 22.5 (23.7) | 25.8 (22.8) |
| Score of total physical activity[a] (SD) | 3.8 (1.9) | 4.0 (1.8) | 4.3 (1.7) | 4.7 (1.8) |
| Total energy intake (kcal/d) | 2 547 (930) | 2 519 (936) | 2 519 (998) | 2636 (1 054) |
| Adherence to Mediterranean diet score[b] (SD) | 5.9 (1.9) | 5.9 (1.8) | 5.8 (1.8) | 5.8 (1.8) |
| Sleep (h/d) | 7.3 (0.9) | 7.3 (0.8) | 7.4 (0.7) | 7.4 (0.7) |
| Smoking, n (%) | | | | |
| Never smoker | 48 (40.3) | 610 (45.7) | 626 (52.1) | 150 (55.4) |
| Current smoker | 30 (25.2) | 322 (24.1) | 242 (20.2) | 51 (18.8) |
| Former smoker | 41 (34.5) | 404 (30.2) | 333 (27.7) | 70 (25.8) |
| Lifetime tobacco exposure (pack-years) (SD) | 8.4 (12.2) | 6.8 (10.1) | 5.1 (8.6) | 4.0 (6.5) |
| Alcohol intake (g/d) (SD) | 6.0 (9.7) | 7.4 (11.2) | 6.5 (8.4) | 7.5 (11.7) |
| Prevalent disease at baseline, n (%) | | | | |
| Depression | 35 (29.4) | 150 (11.2) | 80 (6.6) | 14 (5.1) |
| Diabetes | 3 (2.5) | 20 (1.5) | 9 (0.7) | 1 (0.3) |
| Cancer | 6 (5.0) | 31 (2.3) | 23 (1.9) | 5 (1.8) |
| CVD[c] | 1 (0.8) | 16 (1.2) | 7 (0.5) | 2 (0.7) |
| Body mass index (SD) | 23.9 (3.8) | 23.7 (3.3) | 23.3 (3.2) | 22.6 (2.9) |
| Social life with friends (>3 h/wk) (%) | 92 (77.3) | 994 (74.4) | 939 (78.2) | 215 (79.3) |

[a]Ainsworth, et al. (2011). Compendium of Physical Activities: A Second Update of Codes and MET Values. Medicine & Science in Sports & Exercise 43(8), 1575–1581.

[b]Schröder, et al. (2011). A Short Screener Is Valid for Assessing Mediterranean Diet Adherence among Older Spanish Men and Women. The Journal of Nutrition, 141(6), 1140–1145.

[c]Includes stroke myocardial infarction, CHD, coronary artery surgery or angioplasty

SD: standard deviation, CVD: cardiovascular disease.

## Sensitivity analyses

Results from sensitivity analyses did not substantially change (Table 3). Notably, when adjusting for the change in SRH between baseline and year 4 of follow-up, the associations were even stronger, with participants reporting excellent health showing a 20.4-point higher PWB score (95% CI: 16.4, 24.4) compared to those reporting fair/poor health.

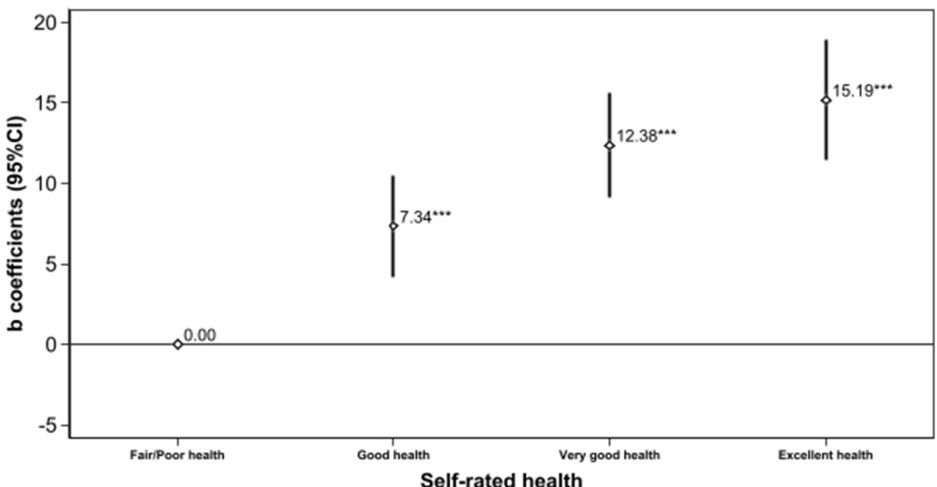

**Fig 3. Adjusted\* beta coefficients of overall psychological well-being after 14-years of follow-up according to categories of self-rated health measured at baseline (n = 2 927, SUN Cohort).** \* Adjusted for sex, age, level of education, marital status, number of children, leisure time physical exercise, a score of total physical activity, caloric intake, adherence to the Mediterranean diet, hours of sleep per day, smoking, lifetime tobacco exposure (pack-years), alcohol consumption, the frequency of interaction in social networks, prevalence of depression, diabetes, cardiovascular disease and cancer and body mass index.

### Longitudinal changes over time

Changes in SRH over the 4-year follow-up period were significantly associated with PWB. Compared to participants whose SRH remained stable (n = 1,704), those who experienced improved SRH (n = 496) showed significantly higher overall PWB scores (β = 5.1; 95% CI: 3.4, 6.8), while those whose SRH worsened (n = 540) showed significantly lower PWB scores (β = −2.6; 95% CI: −4.3, −0.9). This pattern was consistent across most PWB dimensions, with the strongest associations observed for environmental mastery and purpose in life (both β = 1.2 for improved SRH) (Fig 5 and S2 Table).

### Analyses with dichotomized variables

Lastly, PWB score and its six dimensions were analyzed as dichotomous variables using the median as the cut-point. Compared to participants with fair/poor self-rated health, those reporting excellent health had three times higher odds of having above-median overall PWB (OR=3.0; 95% CI: 1.8, 4.8) (Fig 6 and S3 Table). A significant dose-response relationship was observed across SRH categories (p for trend <0.001). Similar patterns were found for all PWB dimensions.

## Discussion

### Summary of main results

Our results showed that university graduates which rated their health as excellent had significantly higher overall PWB than those who reported fair or poor health after a mean follow-up of 14.6 years. This positive association was consistently observed across all six dimensions of PWB: self-acceptance, positive relations with others, autonomy, environmental mastery, purpose in life, and personal growth. Similar results were observed after stratifying the results by sex, age, marital status, education level, and total physical activity. Multiple sensitivity analyses supported the robustness of our

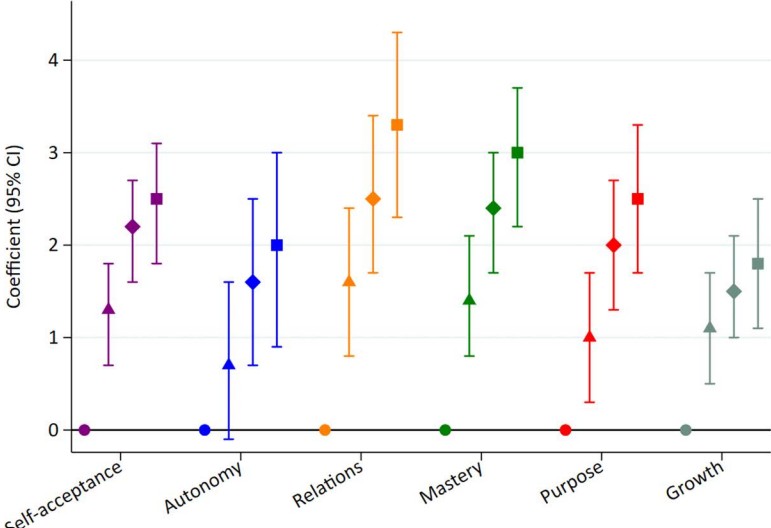

**Fig 4. Adjusted\* beta coefficients of 6 psychological well-being dimensions after 14-years of follow-up according to categories of self-rated health measured at baseline: good, very good and excellent health compared with fair/poor health (n = 2 927, SUN Cohort).** \* Adjusted for sex, age, level of education, marital status, number of children, leisure time physical exercise, a score of total physical activity, caloric intake, adherence to the Mediterranean diet, hours of sleep per day, smoking, lifetime tobacco exposure (pack-years), alcohol consumption, the frequency of interaction in social networks, prevalence of depression, diabetes, cardiovascular disease and cancer and body mass index.

findings. In addition, a 4-year SRH increase was also prospectively associated to a higher PWB, as well as with each one of its dimensions.

## Relation to previous studies

These findings are in line with those of other authors who analyzed the associations between SRH and PWB in cross-sectional studies [25–27]. In a study with older adults in Mongolia, Otgon et al. observed a significant association between a higher SRH and PWB. Findings by Taloyan et al., in a study of Kurdish-born and Swedish-born people, showed that participants with poor SRH had more than a threefold higher odds of having poor PWB compared to those with good SRH. Furthermore, Harschel et al. observed a significant association between SRH and several PWB dimensions including purpose in life, autonomy, and environmental mastery in a sample of Swedish and Latvian elderly. Our study not only corroborates these earlier findings but also expands upon them by demonstrating the long term association between SRH and PWB. Specifically, our analysis shows that a higher baseline SRH was significantly associated to a higher PWB after 14.6 years of follow-up. This longitudinal evidence strengthens the argument that SRH is strongly associated to future PWB, highlighting the critical role of subjective health assessments in forecasting long-term well-being outcomes. While prior studies have primarily examined how PWB predicts subsequent SRH in cross-sectional or short-term designs, our study advances this field by demonstrating the reverse temporal relationship—that SRH strongly predicts PWB over an extended period of time.

## Interpretation of longitudinal association

A particularly novel finding of our study is the significant association between changes in SRH and subsequent PWB. An increased SRH between baseline and year 4 of follow-up, was associated with higher PWB. This dynamic perspective—where changes in perceived health over time influence later PWB adds a new dimension to the understanding of the SRH-PWB relationship. While previous studies have primarily focused on static measures of SRH at a single time point

**Table 2. Multivariable-adjusted beta coefficients and 95% CI of overall psychological well-being at 14-year follow-up, according to self-rated health as a dichotomous variable. Stratified analyses. (SUN Cohort).**

| | N | Fair/ Poor/ Good | Very good/ Excellent | Interaction p-value |
|---|---|---|---|---|
| Sex (95% CI) | | | | 0.629 |
| Female | 1 626 | 0 (ref.) | 6.1 (4.4, 7.9) | |
| Male | 1 301 | 0 (ref.) | 6.0 (4.2, 7.9) | |
| Age (95% CI) | | | | 0.580 |
| <40 years | 1 661 | 0 (ref.) | 6.3 (4.6, 8.0) | |
| ≥40 years | 1 266 | 0 (ref.) | 5.8 (3.9, 7.7) | |
| Years of university education (95% CI) | | | | 0.529 |
| 3–5 years | 2 399 | 0 (ref.) | 6.2 (4.8, 7.6) | |
| 6–9 years | 528 | 0 (ref.) | 5.1 (2.2, 8.0) | |
| Married (95% CI) | | | | 0.568 |
| No | 1 274 | 0 (ref.) | 6.6 (4.6, 8.6) | |
| Yes | 1 653 | 0 (ref.) | 5.5 (3.9, 7.2) | |
| Total physical activity score (95% CI) | | | | 0.567 |
| ≤4 | 1 563 | 0 (ref.) | 6.6 (4.8, 8.4) | |
| >4 | 1 364 | 0 (ref.) | 5.4 (3.6, 7.2) | |

N: number of participants.

Adjusted for sex, age, level of education, marital status, number of children, leisure time physical exercise, a score of total physical activity [48], energy intake, adherence to the Mediterranean diet, hours of sleep per day, smoking, lifetime tobacco exposure (pack-years), alcohol consumption, the frequency of interaction in social networks, prevalence of depression, diabetes, cardiovascular disease and cancer, and body mass index.

**Table 3. Multivariable-adjusted beta coefficients and 95% confidence intervals for psychological well-being according to categories of self-rated health within the 14-year follow-up of the SUN cohort: Sensitivity analysis.**

| | N | Fair or poor | Good | Very good | Excellent | p for trend |
|---|---|---|---|---|---|---|
| Main analysis | 2 927 | 0 (ref.) | 7.3 (4.1,10.5) | 12.3 (9.1,15.6) | 15.1 (11.4,18.8) | <0.001 |
| Excluding prevalent cases of diabetes, CVD and cancer | 2 805 | 0 (ref.) | 7.3 (4.0, 10.6) | 12.4 (9.0, 15.7) | 14.9 (11.0, 18.7) | <0.001 |
| Excluding prevalent cases of depression | 2 648 | 0 (ref.) | 4.7 (1.0, 8.4) | 10.1 (6.4, 13.8) | 12.6 (8.5, 16.7) | <0.001 |
| Excluding all chronic diseases (diabetes, CVD, cancer, and depression) | 2547 | 0 (ref.) | 4.7 (1.0, 8.5) | 10.2 (6.4, 14.0) | 12.4 (8.2, 16.6) | <0.001 |
| PWB transformed with the inverse normal transformation | 2 927 | 0 (ref.) | 0.33 (0.1, 0.5) | 0.62 (0.4, 0.8) | 0.82 (0.6, 1.0) | <0.001 |
| Adjusted for the change in the SRH between the baseline and year 4 of follow up | 2 740 | 0 (ref.) | 9.1 (5.8, 12.4) | 15.8 (12.4, 19.2) | 20.4 (16.4, 24.4) | <0.001 |
| Complete-case analysis | 2 368 | 0 (ref.) | 8.8 (5.3, 12.5) | 14.2 (10.6, 17.9) | 17.4 (13.2, 21.5) | <0.001 |

N: number of participants, CVD: cardiovascular disease, SRH: self-rated health.

Adjusted for sex, age, level of education, marital status, number of children, leisure time physical exercise, a score of total physical activity, caloric intake, adherence to the Mediterranean diet, hours of sleep per day, smoking, lifetime tobacco exposure (pack-years), alcohol consumption, the frequency of interaction in social networks, prevalence of depression, diabetes, cardiovascular disease and cancer and body mass index

[25–27,51,52], our findings suggest that tracking changes in SRH can provide valuable insights into future psychological well-being. This highlights the potential for interventions, such as those focusing on modifiable lifestyle factors, symptom management or social connection, aimed at improving individuals' perceptions of their health to have lasting positive effects on their PWB.

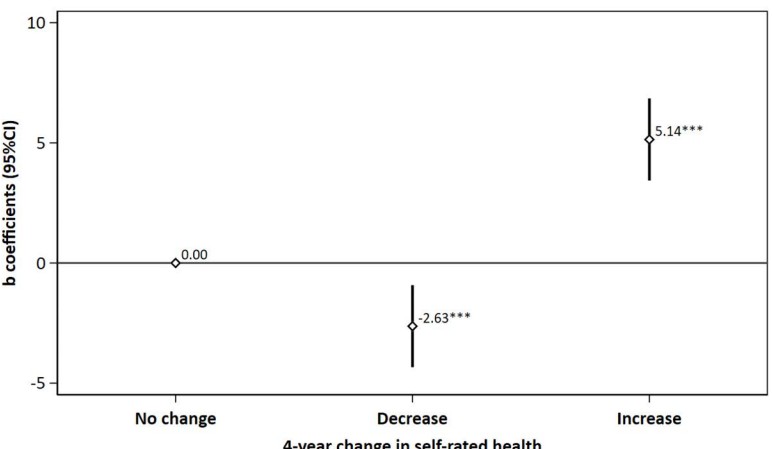

**Fig 5. Adjusted\* beta coefficients of psychological wellbeing (Ryff's score) according to changes in self-rated health (n = 2 740, SUN Cohort).**
\* Adjusted for sex, age, level of education, marital status, number of children, leisure time physical exercise, a score of total physical activity, caloric intake, adherence to the Mediterranean diet, hours of sleep per day, smoking, lifetime tobacco exposure (pack-years), alcohol consumption, the frequency of interaction in social networks, prevalence of depression, diabetes, cardiovascular disease and cancer and body mass index.

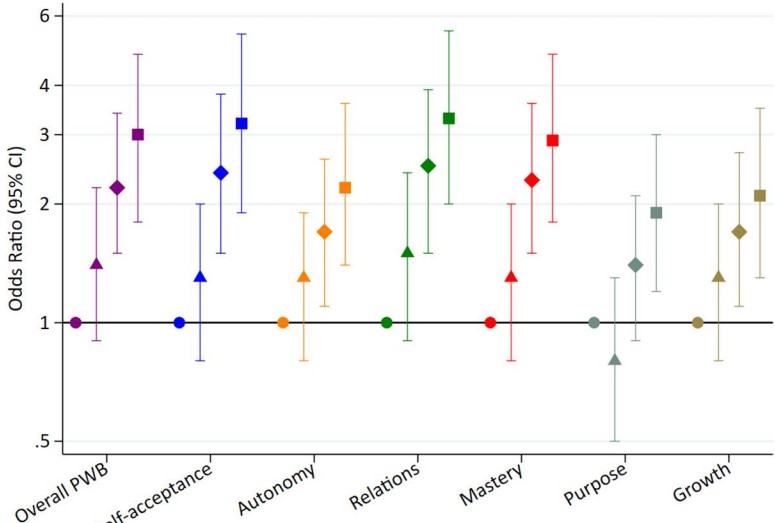

**Fig 6. Adjusted\* odds ratio and 95% CI for above-median psychological well-being dimensions according to baseline self-rated health categories: good, very good and excellent health compared with fair/poor health (SUN cohort).** \* Adjusted for sex, age, level of education, marital status, number of children, leisure time physical exercise, a score of total physical activity, caloric intake, adherence to the Mediterranean diet, hours of sleep per day, smoking, lifetime tobacco exposure (pack-years), alcohol consumption, the frequency of interaction in social networks, prevalence of depression, diabetes, cardiovascular disease and cancer and body mass index.

## Potential mechanisms

Only a few theories have explored the possible mechanisms that could explain the influence of SRH on PWB. In a recent study by Rodríguez-Cifuentes et al., SRH was shown to fully mediate the association between the behavioral component of leisure attitude and PWB, suggesting that individuals with a positive leisure attitude may perceive their

health more favorably, thereby enhancing their PWB [53]. Another possible explanation is that behavioral factors such as higher quality diet and good sleep, correlate with favorable SRH, which in turn supports better psychological outcomes [54].

In contrast to our study, a number of studies have explored the relationship between PWB and posterior SRH in longitudinal designs [23,55]. Our approach uncovers an underexplored dynamic, specifically the possibility that an individual's subjective assessment of their health may indicate which direction the PWB may take over time. This new focus on SRH as a precursor to PWB adds valuable nuance to the bidirectional relationship between the two. These results can have important implications for future interventions. If SRH is found to predict PWB, interventions aimed at improving both objective health and individuals' perceptions of their health—through education, health literacy, and clinical management—could enhance PWB. This could be particularly beneficial in clinical populations where comprehensive health management strategies are needed.

From a clinical practice perspective, routinely assessing SRH with a single question could serve as a practical screening tool in everyday patient encounters. Given that SRH predicts long-term PWB, declining self-rated health may signal the need for comprehensive evaluation of both physical and psychological well-being. This minimal-burden assessment could help clinicians identify patients who may benefit from holistic interventions and timely referral to appropriate support services.

### Strengths and limitations

The strengths of this study include its sample size and prospective nature. The cohort design of the SUN study enabled a long follow-up period (mean of 14.6 years), far exceeding that of many other studies, which are mostly cross-sectional [24–27]. This, in turn, allows for a better comprehension of the relationship between SRH and PWB over time. However, this study also presents some limitations. As with all research based on self-administered questionnaires or self-reported items, there is a risk of information bias, including memory, interpretation, or social desirability biases, which can affect the accuracy of responses and the validity of the results. Nonetheless, the use of validated questionnaires and the highly educated nature of our sample likely mitigated these concerns to some extent. It is important to highlight, though, that SRH is self-reported by nature which is a key strength allowing it to capture personal perceptions beyond objective medical indicators—including social networks, cultural background, socioeconomic status, past health experiences, and personal traits [56,57] – and no differential information bias can be assumed according to the subsequent PWB level. Another limitation is the fact that PWB was only measured once, which restricts our ability to fully understand the stability or changes in PWB over time. This prevents us from examining whether PWB changes mediate the SRH-PWB relationship or whether improvements in SRH correspond to improvements in PWB trajectories. Future research that incorporates longitudinal designs with repeated measures of PWB would provide a more comprehensive understanding of how this aspect fluctuates, develops, and is influenced by various factors over the life course. This approach would offer valuable insights for designing interventions and policies to promote optimal PWB at different life stages. Finally, while the fact that participants are university graduates ensures sample homogeneity and the quality of collected data, and therefore strengthens internal validity, it limits external validity, as the findings cannot be generalized to the broader population. Additional longitudinal research from multiple geographic contexts and with repeated measures of PWB are needed to further explore the underlying mechanisms of this relationship. This includes an assessment of the mediating role of mental illness in the SRH and PWB relationship and identifying other potential mediators such as sociodemographic and economic factors, lifestyle habits and social interactions. A deeper understanding of these factors would allow for more precise health promotion and well-being strategies, as well as more effective preventive interventions. Expanding the sample to include other populations will also help to examine the generalizability of the results.

## Conclusions

In conclusion, this study found a significant association between SRH and subsequent PWB in a cohort of Spanish university graduates. Our findings highlight the temporal relationship between SRH and various dimensions of PWB, while adjusting for a broad range of sociodemographic, health and lifestyle variables. These results emphasize the importance of a holistic approach to health promotion, disease prevention, and health management, one that addresses both the physical health and PWB.

## Supporting information

**S1 Table. Multivariable-adjusted beta coefficients and 95% confidence intervals for associations between baseline self-rated health and psychological well-being after 14 years of follow-up (n = 2,927, SUN cohort).**
(DOCX)

**S2 Table. Beta coefficients and 95% CI of overall psychological well-being and 6 dimensions according to the change in the SRH between the baseline and 4 years of follow up (N = 2 740).**
(DOCX)

**S3 Table. Multivariable adjusted\* odds ratio and 95% CI for above-median psychological well-being (and its six dimensions) after 14 years according to baseline self-rated health categories (SUN cohort).**
(DOCX)

## Author contributions

**Conceptualization:** Maria Vasilj, Miguel Ruiz-Canela.

**Data curation:** Leticia Goni, Miguel Ruiz-Canela.

**Formal analysis:** Maria Vasilj, Leticia Goni, Miguel Ruiz-Canela.

**Writing – original draft:** Maria Vasilj.

**Writing – review & editing:** Estefanía Toledo, Prado Silván-Ferrero, Encarnación Nouvilas-Pallejá, Alessandra Jarufe, Leticia Goni, Maira Bes-Rastrollo, Miguel Ruiz-Canela.

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
