## [Decision Letter · Decision Letter 0]

8 Sep 2025

PONE-D-25-26964Longitudinal association between self-rated health and psychological well-being in a sample of Spanish university graduatesPLOS ONE

Dear Dr. Ruiz-Canela,

Thank you for submitting your manuscript to PLOS ONE. After careful consideration, we feel that it has merit but does not fully meet PLOS ONE’s publication criteria as it currently stands. Therefore, we invite you to submit a revised version of the manuscript that addresses the points raised during the review process.

We look forward to receiving your revised manuscript.

Kind regards,

Boshra A. Arnout, Professor

Academic Editor

PLOS ONE

**Journal Requirements:**

1. When submitting your revision, we need you to address these additional requirements. Please ensure that your manuscript meets PLOS ONE's style requirements, including those for file naming. The PLOS ONE style templates can be found at https://journals.plos.org/plosone/s/file?id=wjVg/PLOSOne_formatting_sample_main_body.pdf and https://journals.plos.org/plosone/s/file?id=ba62/PLOSOne_formatting_sample_title_authors_affiliations.pdf 2. Please note that your Data Availability Statement is currently missing the repository name and/or the DOI/accession number of each dataset and the direct link to access each database. If your manuscript is accepted for publication, you will be asked to provide these details on a very short timeline. We therefore suggest that you provide this information now, though we will not hold up the peer review process if you are unable. 3. Please include captions for your Supporting Information files at the end of your manuscript, and update any in-text citations to match accordingly. Please see our Supporting Information guidelines for more information: http://journals.plos.org/plosone/s/supporting-information. 4. If the reviewer comments include a recommendation to cite specific previously published works, please review and evaluate these publications to determine whether they are relevant and should be cited. There is no requirement to cite these works unless the editor has indicated otherwise. 

Reviewers' comments:

Reviewer's Responses to Questions

**Comments to the Author**

1. Is the manuscript technically sound, and do the data support the conclusions?

Reviewer #1: Yes

Reviewer #2: Partly

2. Has the statistical analysis been performed appropriately and rigorously?

Reviewer #1: Yes

Reviewer #2: I Don't Know

3. Have the authors made all data underlying the findings in their manuscript fully available?

Reviewer #1: Yes

Reviewer #2: Yes

4. Is the manuscript presented in an intelligible fashion and written in standard English?

Reviewer #1: Yes

Reviewer #2: Yes

5. Review Comments to the Author

**Reviewer #1: **This study examined the longitudinal association between self-rated health and subsequent psychological well-being in a Spanish cohort. The study pertains to an important area. My observations are given below:

(1) The study title is okay.

(2) The abstract needs revision. The goal, methodology, and main conclusions of the study are succinctly and clearly summarized in the abstract. A more succinct conclusion and greater focus on limitations would be beneficial, though.

(3) The introduction also needs minor revision. The well-organized introduction clearly explains the study's purpose by pointing out a pertinent gap in the body of existing literature. It successfully sets apart earlier studies that looked at PWB as an indicator of SRH and emphasizes the necessity of investigating the opposite relationship. Instead of being listed in order, the review of earlier research could be more critically synthesized. Its coherence and scholarly impact would be improved by more precisely stating the longitudinal approach's novelty and highlighting its theoretical foundations sooner. Additionally, minor redundancies ought to be eliminated.

(4) The method again needs minor amendments. The methods section, which covers design, sampling, variables, and analysis procedures, is thorough and organized. Strength is increased by using validated tools and longitudinal data. However, bias may be introduced if self-report questionnaires are used, particularly for SRH and PWB. There is no rationale for handling missing data, especially assumptions about smoking and the number of children, and doing so could jeopardize validity. Additionally, more information regarding participant attrition and retention is required. Nevertheless, the study's methodological rigor is improved by the strong analytical approach, which includes stratified analyses and sensitivity.

(5) The results need modifications. A wide variety of covariates are included in the results section, indicating a comprehensive statistical approach. Interpretability is, however, constrained by the ambiguity of the reporting of particular findings, effect sizes, and significance levels. It seems as though the text is disjointed and unrelated to the research questions. Important psychological variable outcomes are not given enough attention, and it's unclear how changes affected the main findings. The findings would be more transparent, coherent, and impactful overall if the data were presented more clearly and backed up by pertinent tables or figures.

(6) The discussion needs minor improvements. The findings are thoroughly synthesized in the discussion, which also highlights the novel contribution of tracking SRH changes over time and makes good connections with previous literature. Insightful explanations of the mechanisms and intervention implications are provided. The section, though, could use more organization and is far too long. Although the limitations are discussed, a more critical discussion of the use of self-report and the lack of repeated PWB measures would be beneficial. The impact and clarity of the discussion would be further enhanced by making clear how the findings go beyond earlier research.

**Reviewer #2:** Review of Longitudinal association between self-rated health and psychological well-being in a sample of Spanish university graduates

Thank you for an interesting manuscript describing the association between PWB and SRH and the change of SRH within 4 years, and the association with PWB, using a large sample with a long follow-up.

Overall, it is well-written, but I have some small comments listed under each heading.

A major comment

All tables should be able to stand alone, with only the table text included. I think none of the tables can do so at the moment. Currently, I am unable to interpret the tables without going back to the method and statistics and reading once again what analysis you present the result of in each table.

I think you should choose one thing to analyze, either the change in SRH over the 4 years or the baseline SRH, to make it easier to interpret.

Abstract

Generally, I get a bit disturbed over the use of abbreviations within the abstract. Personally, I think it lowers the readability of the text. Please review to determine if abbreviations are necessary to avoid exceeding the word and character limitations.

The description of the participants' age is the age at baseline or at follow-up; please clarify.

Introduction

Row 65: Why do you have a secondary objective? Is there any reason to believe that SRH should change in 4 years? Please clarify what your basis is for this idea. You haven’t mentioned anything about a change in SRH over time or what events might influence it. I think there are some studies done describing changes in SRH over time, which could be good to incorporate.

Methods

Write out the abbreviation first time used, SUN, which I presume you mean Seguimiento Universidad de Navarra

Even if the study has been described elsewhere, I still want to know how long the inclusion of participants was ongoing. How many were included at baseline, proportion of men and women.

Was it decided at study start that the follow-up questionnaire should be filled out after 14 years?

If I understand it right, you included participants from 1999 and 2022? Or between which time points were the included participants recruited?

Figure 1.

What is included in pre-baseline variables? Was the pre-baseline 4 years before? What information is gathered from pre-baseline? I think you need to add some text to explain Figure 1to guide the reader. A figure or a table should be able to stand alone and be understood.

Participants

Row 85, Follow-up from which time point?

Row 86, which time point (t?) do you mean is the baseline in this context?

Figure 2 is very nice and easy to understand.

Row 90, when did the participants receive the written information?

I am missing the age at which they were included in the study.

Variables

Rows 103- 105, I think you should rewrite these sentences to make it easier to understand. Especially the sentence starting with “It has been…”

For example,” The latter item served as an indicator of how frequently participants interacted within their social networks and has been widely applied in epidemiological studies examining associations …..…”

Statistics

Row 126, I am unfamiliar with the wording raw model, In previous papers, I think I have seen it mentioned as a crude model, but it may be the same, but different wording

You have used a lot of different variables in the multiple linear regression. Did you analyse whether they were correlated with each other?

I think that METS/ week and a score of total physical activity are strongly correlated and may not be proper to use both. The same goes for smoking habits and lifetime tobacco exposure

There appear to be several potential limitations in the statistical analysis, as various variables may act as mediators or confounders of the outcome. I cannot see that you have investigated VIF (Variance Inflation Factor) to detect multicollinearity in the regression models.

Why did you adjust for depression, diabetes, cardiovascular disease, and cancer instead of excluding them from the analysis, since it is well-known that a chronic disease affects SRH and the well-being, alternatively conduct a separate analysis for the group with chronic disease?

Due to the advanced statistical analysis, I would recommend that the analysis be reviewed by an expert in statistics.

Rows 155-160, it is nicely described how you have handled missing values. However, I did not think this was the right way; a more proper approach would be to exclude those who didn´t answer, just because you do not know the reason why they didn’t answer. I could be due to a lack of time. Smokers could be ashamed to admit that they are smokers and, because of this, skip answering the question, or have you thought about the method of carrying the last number forward? Regarding marital status, I think that was a proper way.

Please explain how you did the imputation in hours of sleep; the way you have done it seems risky and may affect the results.

Results

Row 164, follow-up not followed-up.

Table 1. There is a lack of information about what the numbers in the parentheses represent in several of the variables.

Row 182, I think it should be more proper to present the main findings of Table 2 and then refer to the table, instead of writing “Table 2 presents the….”

Table 2, first row (mean, SD) in parentheses, is it the mean or SD?

Row 211, see the comment above regarding row 182. And the table shows the result of the multiple linear regression stratified by…..

Row 211, since I haven't seen any explanation of whether the age refers to baseline, pre-baseline, or follow-up, it is hard to interpret

Tables 3, 4, and 5: What do the numbers within the parentheses represent?

Discussion

Row 267, If this is the particular novel finding I think this should be presented as the main finding. As you write in rows above, your findings just (mor or less) replicate previous findings.

Row 273, What kind of intervention do you imagine may affect the individual's perception of their health? Is it just their perceptions that should be changed? Or is it the actual physical health that needs improvement for a changed perception of one´s own health?

Row 282, is it really the subjective assessment that shapes the PWB over time? I think it is more complex. The measurement may indicate in which direction the PWB may take over time (increasing or decreasing)

Row 283- 287, Can you elaborate? It sounds like you mean that it is only the attitude to how you perceive your health that should change, and the actual physical and mental health, and further aspects do not affect the perception of one´s health.

How do you think your results should be used clinically? Yes, I understand that a holistic approach to health promotion and prevention is necessary, but in what specific way should I use it in my everyday work?

6. PLOS authors have the option to publish the peer review history of their article (what does this mean?). If published, this will include your full peer review and any attached files.

Reviewer #1: **Yes: **Gyanesh Kumar Tiwari

Reviewer #2: No

---

## [Author Response · Author response to Decision Letter 1]

28 Oct 2025

Reviewer #1: This study examined the longitudinal association between self-rated health and subsequent psychological well-being in a Spanish cohort. The study pertains to an important area. My observations are given below:

RESPONSE: Thank you for your review. We believe all your comments have been very helpful to improve the manuscript.

(1) The study title is okay.

(2) The abstract needs revision. The goal, methodology, and main conclusions of the study are succinctly and clearly summarized in the abstract. A more succinct conclusion and greater focus on limitations would be beneficial, though.

RESPONSE: Thank you for your comment. We have modified the abstract to improve the conclusion and acknowledge the main limitations.

Initial version: “Conclusions: SRH showed direct, long-term association with subsequent PWB among Spanish university graduates, suggesting that perceived health status may be an important determinant of future PWB. These findings have practical implications for well-being programs and warrant future research in diverse populations and on underlying mechanisms.”

New version (rows 35-38): “Conclusion: Self-rated health showed a direct, long-term association with subsequent PWB among Spanish university graduates, suggesting that perceived health status may be an important determinant of future PWB. However, our findings are limited by the single measurement of PWB and the use of self-reported measures. Further studies in more diverse populations are warranted to confirm these results.”

(3) The introduction also needs minor revision. The well-organized introduction clearly explains the study's purpose by pointing out a pertinent gap in the body of existing literature. It successfully sets apart earlier studies that looked at PWB as an indicator of SRH and emphasizes the necessity of investigating the opposite relationship. Instead of being listed in order, the review of earlier research could be more critically synthesized. Its coherence and scholarly impact would be improved by more precisely stating the longitudinal approach's novelty and highlighting its theoretical foundations sooner. Additionally, minor redundancies ought to be eliminated.

RESPONSE: We have incorporated your suggestions and modified the Introduction section as follows:

Initial version: “However, there is limited research that looks into the relationship between the SRH and PWB [22–28]. Taloyan et al. [26] found a moderate association between PWB and SRH in a cross-sectional study, and Ryff et al. observed PWB as a predictor of better self-rated health [23]. Choi & Jung’s (2025) machine learning analyses revealed that psychological well-being, personality traits, and emotional states strongly predict self-rated health[28]. All three of these studies looked at the impact that PWB might have on SRH [23,24]; however, there is little evidence on whether the relationship holds true in reverse. Therefore, more research focusing on the prospective association between SRH and subsequent PWB is required to better understand a possible bidirectional relationship between SRH and PWB.”

New version (rows 50-60): “However, research examining the relationship between SRH and PWB remains limited and methodologically constrained [22–28]. Existing studies are predominantly cross-sectional [22,24–27] or feature short follow-up periods, limiting causal inference. Moreover, available longitudinal research has primarily investigated PWB as a predictor of subsequent SRH [23,28], with Ryff et al. demonstrating that higher PWB predicted better self-rated health over time [23], and Choi & Jung's machine learning analyses identified PWB as a strong predictor of SRH [28]. This unidirectional focus leaves a significant gap in understanding whether SRH might similarly influence future PWB. More critically, this relationship remains largely unexplored in longitudinal designs. Therefore, investigating the prospective association between SRH and subsequent PWB over an extended follow-up period is essential to gain a more comprehensive understanding of the bidirectional relationship between these constructs.”

Additional modifications to the Introduction section were made in response to suggestions from the other reviewer. Specifically, the reviewer requested justification for distinguishing between primary and secondary objectives, prompting the addition of several phrases and supporting citations:

22.Diržytė A, Perminas A. Self-reported health-related experiences, psychological capital, and psychological wellbeing in Lithuanian adults sample. Health Psychol Open. 2021;8. doi:10.1177/2055102921996164

23. Ryff CD, Radler BT, Friedman EM. Persistent Psychological Well-being Predicts Improved Self-Rated Health Over 9-10 Years: Longitudinal Evidence from MIDUS. Health Psychol Open. 2015;2. doi:10.1177/2055102915601582

24. Piko B. Health-related predictors of self-perceived health in a student population: the importance of physical activity. J Community Health. 2000;25: 125–137. doi:10.1023/A:1005129707550

25. Otgon S, Myagmarjav S, Burnette D, Lkhagvasuren K, Casati F. Sociodemographic predictors of flourishing among older adults in rural and urban Mongolia. Sci Rep. 2023;13. doi:10.1038/S41598-023-28791-X

26. Taloyan M, Sundquist J, Al-Windi A. The impact of ethnicity and self-reported health on psychological well-being: A comparative study of Kurdish-born and Swedish-born people. Nord J Psychiatry. 2008;62: 392–398. doi:10.1080/08039480801984263

27. Harschel AK, Schaap LA, Iwarsson S, Horstmann V, Tomsone S. Self-rated health among very old people in European countries: An explorative study in Latvia and Sweden. Gerontol Geriatr Med. 2015. doi:10.1177/2333721415598432

28. Choi JH, Jung DH. The role of psychological factors in predicting self-rated health: implications from machine learning models. Psychol Health Med. 2025 [cited 11 Mar 2025]. doi:10.1080/13548506.2025.2450546

(4) The method again needs minor amendments. The methods section, which covers design, sampling, variables, and analysis procedures, is thorough and organized. Strength is increased by using validated tools and longitudinal data. However, bias may be introduced if self-report questionnaires are used, particularly for SRH and PWB.

RESPONSE: We appreciate the reviewer’s concern regarding the use of self-reported data, which may indeed introduce some degree of imprecision. To address this limitation, we used validated questionnaires, and we believe that the fact that participants were university graduates likely enhanced the reliability and quality of the self-reported information. Moreover, in the case of SRH, one of the main strengths of this measure lies precisely in its self-reported nature. Previous research has shown that responses to this simple question capture a wide range of personal and contextual factors that shape how individuals perceive and evaluate their own health (Jylhä, 2009).

Jylhä M. What is self-rated health and why does it predict mortality? Towards a unified conceptual model. Soc Sci Med. 2009;69: 307–316. doi:10.1016/J.SOCSCIMED.2009.05.013

Row 122:

We have added the following in Methods:

“Although self-report questionnaires may introduce potential bias, the validated nature of our instruments and the high educational level of participants likely enhanced response reliability and data quality.”

And also in the discussion we have acknowledged the limitation of using self-reported questionnaires:

Row 327:

“Nonetheless, the use of validated questionnaires and the highly educated nature of our sample likely mitigated these concerns to some extent.”

There is no rationale for handling missing data, especially assumptions about smoking and the number of children, and doing so could jeopardize validity.

RESPONSE: We thank the reviewer for this important comment. We have clarified our rationale for handling missingness in smoking status and number of children, explaining that these cases were assumed to be missing not at random because participants often skip such questions when they perceive them as not applicable (e.g., non-smokers or individuals without children). To address concerns about potential bias, we also performed a complete-case analysis, which yielded results consistent with the main analysis. This strengthens confidence that our assumptions did not materially affect study conclusions.

Initial version: “In the cases of missing number of children and smoking status, we assumed that the answer was negative, meaning that participants did not answer because they understood that the question did not apply to them (don’t have children or never smoked). For marital status, we categorized those participants with a missing value as “Other”. For hours of sleep, we performed a simple imputation where PWB, SRH, sex, age, physical activity and total physical activity score were used as predictors.”

New version (rows 172-182): “We assumed that missingness in smoking status and number of children was not random. Specifically, when participants do not provide information on smoking or children, it is often because they perceive the question as not applicable to them. This type of non-response is consistent with a missing not at random mechanism, as the probability of missingness is directly related to the unreported value itself (e.g., a non-smoker omits smoking status, or a participant without children omits number of children). Therefore, we coded missing values in smoking as 'never smoked,' and missing values in number of children as 'none.' For marital status, we categorized missing values as 'Other.' For hours of sleep, which showed the highest missing proportion, we used regression imputation [50] with predictors including psychological well-being (PWB), self-rated health (SRH), sex, age, physical activity, and total physical activity score. To ensure that these assumptions did not bias the findings, we additionally performed a complete-case analysis excluding individuals with missing values.”

50. Kang H. The prevention and handling of the missing data. Korean J Anesthesiol. 2013;64: 402. doi:10.4097/KJAE.2013.64.5.402

Additionally, more information regarding participant attrition and retention is required. Nevertheless, the study's methodological rigor is improved by the strong analytical approach, which includes stratified analyses and sensitivity.

RESPONSE: Thank you for raising this important question. We have added a citation to the most recent SUN cohort profile (De La Fuente-Arrillaga, 2025), which offers updated and comprehensive information on participant recruitment and follow-up. As mentioned in the cohort profile article, “The overall response rate (attrition) in the SUN cohort was 10%”.

To further clarify this point we updated the Figure 2. Flowchart of inclusion criteria with a more detailed information.

*Some participants skipped certain questionnaires but completed later follow-up assessments. These participants are not considered lost to follow-up, instead they exhibit inconsistent response patterns throughout the study period

De La Fuente-Arrillaga C, Carlos S, Toledo E, Barbería-Latasa M, Razquin C, Ruiz-Canela M, et al. Cohort Profile Update: The ‘Seguimiento Universidad de Navarra’ (SUN) study after 24 years of follow-up. Int J Epidemiol. 2025;54: 149. doi:10.1093/IJE/DYAF149

(5) The results need modifications. A wide variety of covariates are included in the results section, indicating a comprehensive statistical approach. Interpretability is, however, constrained by the ambiguity of the reporting of particular findings, effect sizes, and significance levels. It seems as though the text is disjointed and unrelated to the research questions. Important psychological variable outcomes are not given enough attention, and it's unclear how changes affected the main findings. The findings would be more transparent, coherent, and impactful overall if the data were presented more clearly and backed up by pertinent tables or figures.

RESPONSE: We thank the reviewer for their valuable suggestions, which have helped us improve the clarity and coherence of our Results section. We have undertaken significant revisions to address each concern:

1. Regarding structural coherence and clarity: We have completely restructured the Results section by incorporating clear subtitles that directly align with our research questions and statistical approach. The new organization follows a logical progression:

• Baseline characteristics and descriptive statistics (Table 1)

• Main longitudinal associations between SRH and overall PWB (now Figures 3 and 4, previously Table 2)

• Stratified analyses by key demographic variables (now Table 2)

• Sensitivity analyses (now Table 3)

• Changes in SRH and their association with PWB (now Figure 5)

• Logistic regression results for dichotomous outcomes (now Figure 6 and Supplementary Table 3)

2. Regarding emphasis on psychological outcomes: We have enhanced the presentation of findings related to all six PWB dimensions (self-acceptance, positive relations with others, autonomy, environmental mastery, purpose in life, and personal growth). Each dimension is now reported both in Figure 4 and in Supplementary Table 1.

3. Regarding visualization of key findings: Following the reviewer's suggestion, we have replaced the previous tabular presentation of the main findings with Figure 3. We also substituted the table showing the results of the analysis with dichotomized variables with a figure (Figure 6). Additionally, Supplementary Tables 1-3 provide the detailed numerical data for readers interested in the complete statistical information.

All changes are tracked in the revised manuscript for the reviewer's convenience.

(6) The discussion needs minor improvements. The findings are thoroughly synthesized in the discussion, which also highlights the novel contribution of tracking SRH changes over time and makes good connections with previous literature. Insightful explanations of the mechanisms and intervention implications are provided. The section, though, could use more organization and is far too long. Although the limitations are discussed, a more critical discussion of the use of self-report and the lack of repeated PWB measures would be beneficial. The impact and clarity of the discussion would be further enhanced by making clear how the findings go beyond earlier research.

RESPONSE: Thank you for this constructive feedback. We have made the following improvements to the discussion section:

We incorporated subtitles to the Discussion section to provide better organization and reduce the perceived length through clearer sectioning. These sections are: Summary of main results, Relation to previous studies, Interpretation of longitudinal association, Potential mechanisms, Strengths and limitations and Conclusions.

We acknowledge the reviewer's concern about self-report measures. SRH is inherently self-rated by design, representing a key strength: it encompasses personal perceptions that extend beyond objective medical indicators, including social networks, cultural background, socioeconomic status, past health experiences, and personal traits (Craig et al., 2018; Friedman & Kern, 2014; Read et al., 2016; Roudijk et al., 2017;). Similarly, PWB by definition requires subjective self-assessment. We have modified the following sentence to clarify this relevant question:

Craig, B. A., Morton, D. P., Morey, P. J., Kent, L. M., Gane, A. B., Butler, T. L., Rankin, P. M., & Price, K. R. (2018). The association between self-rated health and social environments, health behaviors and health outcomes: a structural equation analysis. BMC Public Health, 18(1). https://doi.org/10.1186/S12889-018-5323-Y

Friedman, H. S., & Kern, M. L. (2014). Personality, well-being, and health. Annual Review of Psychology, 65, 719–742. https://doi.org/10.1146/annurev-psych-010213-115123

Read, S., Grundy, E., & Foverskov, E. (2016). Socio-economic position and subjective health and well-being among older people in Europe: A systematic narrative review. Aging and Mental Health, 20(5), 529–542. https://doi.org/10.1080/13607863.2015.1023766

Roudijk, B., Donders, R., & Stalmeier, P. (2017). Cultural values: can they explain self-reported health? Quality of L

---

## [Decision Letter · Decision Letter 1]

21 Nov 2025

Longitudinal association between self-rated health and psychological well-being in a sample of Spanish university graduates

PONE-D-25-26964R1

Dear Dr. Ruiz-Canela,

We’re pleased to inform you that your manuscript has been judged scientifically suitable for publication and will be formally accepted for publication once it meets all outstanding technical requirements.

Kind regards,

Boshra A. Arnout, Professor

Academic Editor

PLOS ONE

Additional Editor Comments (optional):

Reviewers' comments:

Reviewer's Responses to Questions

**Comments to the Author**

1. If the authors have adequately addressed your comments raised in a previous round of review and you feel that this manuscript is now acceptable for publication, you may indicate that here to bypass the “Comments to the Author” section, enter your conflict of interest statement in the “Confidential to Editor” section, and submit your "Accept" recommendation.

Reviewer #1: All comments have been addressed

Reviewer #2: All comments have been addressed

2. Is the manuscript technically sound, and do the data support the conclusions?

Reviewer #1: Yes

Reviewer #2: Yes

3. Has the statistical analysis been performed appropriately and rigorously?

Reviewer #1: Yes

Reviewer #2: I Don't Know

4. Have the authors made all data underlying the findings in their manuscript fully available?

Reviewer #1: Yes

Reviewer #2: Yes

5. Is the manuscript presented in an intelligible fashion and written in standard English?

Reviewer #1: Yes

Reviewer #2: Yes

6. Review Comments to the Author

Reviewer #1: I have thoroughly read the revied and updated manuscript and the authors' replies to the reviewers' queries. The authors have adequately addressed all the issues raised, and I am happy with how adequate and clear their revisions are. The manuscript exhibits good methodological rigour, is pertinent to the journal's focus, and has the potential to advance knowledge and contribute to the existing literature.

Well done. No furtehr chanfges are required

Reviewer #2: Thank you for an interesting paper, investigating self rated health and how it is associated with incident disease

7. PLOS authors have the option to publish the peer review history of their article (what does this mean?). If published, this will include your full peer review and any attached files.

Reviewer #1: **Yes: **Gyanesh Kumar Tiwari

Reviewer #2: No

---

## [Editor Report · Acceptance letter]

PONE-D-25-26964R1

PLOS One

Dear Dr. Ruiz-Canela,

I'm pleased to inform you that your manuscript has been deemed suitable for publication in PLOS One. Congratulations! Your manuscript is now being handed over to our production team.

Kind regards,

on behalf of

Professor Boshra A. Arnout

Academic Editor

PLOS One